# Short Peptides Induce Development of Root Hair *Nicotiana tabacum*

**DOI:** 10.3390/plants11070852

**Published:** 2022-03-23

**Authors:** Larisa I. Fedoreyeva, Inna A. Chaban, Neonila V. Kononenko

**Affiliations:** All-Russia Research Institute of Agricultural Biotechnology, Timiryazevskaya 42, 127550 Moscow, Russia; iab@iab.ac.ru (I.A.C.); nilva@mail.ru (N.V.K.)

**Keywords:** root hairs, short peptides, *Nicotiana tabacum* L.

## Abstract

Root hairs absorb soil nutrients and water, and anchor the plant in the soil. Treatment of tobacco (*Nicotiana tabacum*) roots with glycine (Gly) amino acid, and glycilglycine (GlyGly) and glycilaspartic acid (GlyAsp) dipeptides (10^−7^ M concentration) significantly increased the development of root hairs. In the root, peptide accumulation was tissue-specific, with predominant localization to the root cap, meristem, elongation zone, and absorption zone. Peptides penetrated the epidermal and cortical cell and showed greater localization to the nucleus than to the cytoplasm. Compared with the control, tobacco plants grown in the presence of Gly, GlyGly, and GlyAsp exhibited the activation of *WER*, *CPC*, *bHLH54*, and *bHLH66* genes and suppression of *GTL1* and *GL2* genes during root hair lengthening. Although Gly, GlyGly, and GlyAsp have a similar structure, the mechanism of regulation of root hair growth in each case were different, and these differences are most likely due to the fact that neutral Gly and GlyGly and negatively charged GlyAsp bind to different motives of functionally important proteins. Short peptides site-specifically interact with DNA, and histones. The molecular mechanisms underlying the effect of exogenous peptides on cellular processes remain unclear. Since these compounds acted at low concentrations, gene expression regulation by short peptides is most likely of epigenetic nature.

## 1. Introduction

Root hairs play diverse roles that are critical for plant growth and development: absorption of sufficient amounts of water and nutrients from the rhizosphere under drought and/or nutrient deficient conditions; interaction between the plant and soil microfauna; and anchoring the plant in the soil [1,2,3]. The rhizosphere is formed as a result of the proliferation of root hairs that entangle soil particles, forming root sheaths [4]. Increase in the number and length of root hairs could be used to increase crop productivity under stress conditions and improve plant adaptation to nutrient deficient soils [5].

Plant cell differentiation is a complex process [6]. Based on the information gained from genetic and molecular studies conducted in *Arabidopsis thaliana*, root hair development is broadly divided into four stages: root hair cell specification, initiation, elongation (tip growth) and maturation [7,8]. Depending on the type of plant, root hairs are formed from root epidermal cells either randomly or via asymmetric cell differentiation. In the random process, any epidermal cell can develop into a root hair [9,10]. In the asymmetric cell differentiation process, root hairs are produced only by H hair cells (trichoblasts), which are formed from epidermal stem cells during the later period of cell division [11,12]; the other epidermal cells formed from epidermal stem cells develop into normal epidermal cells called non-hair N cells (atrichoblasts) [13]. The fate of epidermal stem cells depends on their position and is controlled by the exchange of signals between adjacent root epidermal and cortical cells and between H and N cells [14,15]. In some cases, N cells can also be transformed into H cells, which produce ectopic root hair cells [13,16].

During root hair elongation, cells in root hair tips exhibit a unique growth pattern, which is regulated by the cytoskeleton and is accompanied by changes in the cell wall [17]. Root hair differentiation involves a variety of cellular and genetic processes [18,19]. The initiation of root hair growth is regulated by a genetic program that determines cell specification [20,21]. Key regulators that translate these developmental signals to initiate root hair development include the basic helix loop helix (*bHLH*) transcription factor ROOT HAIR DEFECTIVE 6 (*RHD6*) and its close homolog RHD6-LIKE 1 (*RSL1*) [22,23].

The *bHLH* superfamily consists of a large number of proteins found virtually in all organisms, including fungi, plants, and animals [24]. The *bHLH* transcription factors act as a link between the R2R3-MYB proteins, such as WEREWOLF (*WER*; which controls the specification of hair root cells) and GLABRA1 (*GL1*; which regulates trichome differentiation), and the WD-40 transcription factor TRANSPARENT TESTA GLABRA1 (*TTG1*) to form an activation complex [25,26,27,28].

Previous studies have shown that the initiation of root hair differentiation is determined by interactions among six genes: *CPC*, *ETC*, *GL2*, *GL3/EGL3*, *TTG*, and *WER* [29]. Among these genes, *CPC*, *GL2*, and *WER* are preferentially expressed in N cells during seed germination, and regulate the expression of *GL3/EGL2* in H cells [30]. Initiation of the differentiation of epidermal cells N to H depends on the sensitivity to signals from cortical cells, and is directly involved in the regulation of *CPC*, *GL2*, and *WER* expression. Interaction between these regulatory genes has been demonstrated previously [31]. At position N, *WER* positively regulates *CPC* expression and *GL2* appears to move into cells at position H and inhibit expression *WER* and *CPC*, which causes the cell to initiate hair formation.

Since the *GL2* gene is predominantly expressed in N cells, it is usually considered a negative regulator of root hair development, and its activation or inhibition determines the fate of root hair development. Therefore, *GL2* is believed to represent an important genetic switch in determining the cell fate during the formation of root hairs [32].

An important question is how plants determine the final size of growing cells. The RSL4 transcription factor promotes the growth of root hairs [33]. Another trihelix transcription factor GT2-LIKE1 (GTL1) and its homolog *DF1* inhibit the growth of hair roots in Arabidopsis. Both GTL1 and DF1 directly bind to the *RSL4* gene promoter and regulate its expression, suppressing the growth of root hairs. Thus, this study reveals a basic regulatory module that precisely determines the degree of root hair growth through the orchestrated actions of opposing transcription factors [34].

Root hair growth is fine-tuned by various hormonal and environmental signals. Auxin is well known to enhance the growth of root hairs [35,36]. Exogenous application of ethylene also promotes root hair growth, and this physiological response is accompanied by the regulation of *RSL4*. In addition, environmental factors, such as nutrient availability, also affect root hair development. It has been suggested that hormones and environmental factors influence root hair initiation in a different manner than that occurring during normal development [37]. Together, these studies suggest that root hair growth is regulated at several levels, although the precise molecular details of this control remain unknown.

Endogenous peptides, like phytohormones, play important roles in the regulation of numerous intercellular connections and physiological activities, and respond to various influences [38]. Peptides interact with signaling phytohormones and regulate a wide range of biological processes. In plants and animals, short peptides induce the expression of transcription factor-encoding genes involved in cell differentiation and plant growth and development [39,40]. Currently, CLE peptides (CLAVATA3/Embryo Surrounding Region-Related) are the most studied secreted peptides in plants [38,40]. These peptides interact with signaling phytohormones and are involved in regulation with the environment, modulating a wide range of biological processes. It was found that CLE peptides were involved in the regulation of seed development, the formation of vessels and lateral roots, and stem cell homeostasis in the apical meristem of seedlings and roots [41,42].

We have previously shown that dipeptides GlyGly and GlyAsp and the amino acid Gly at a concentration of 10^−7^ M in the medium significantly stimulate the growth and development of seedlings of *Nicotiana tabacum*. They stimulate the formation and growth of leaves and roots. Peptides modulate expression of genes of the KNOX and GRF families that are responsible for cell differentiation and encoding transcription factors [43]. The profiles of induction or repression of gene expression by the same peptide in tobacco seedlings are different. This indicates cellular (tissue) specificity of the activity of peptides in plants.

The aim of this study was to investigate the effect of glycine (Gly) amino acid and GlyGly and Gly-aspartic acid (GlyAsp) dipeptides on the regulation of root hair growth in tobacco (*Nicotiana tabacum* L.).

## 2. Results

### 2.1. Biometric Parameters

The amino acid Gly and the dipeptides GlyGly and GlyAsp significantly stimulated the growth and development of tobacco seedlings. GlyAsp and GlyGly showed the most pronounced stimulating, which significantly accelerated the development of seedlings and the root system (Figure 1). The length of the main root was approximately 40% greater in the presence of dipeptides compared with the control (Figure 1). Although the length of the main root increased significantly under the influence of GlyAsp and GlyGly, the total length of the seedling increased only by ~15%. Notably, in the presence of dipeptides, seedlings produced new lateral roots, increasing the adsorption surface (Figure 1). Additionally, the biomass of tobacco seedlings increased by approximately 40% in the presence of dipeptides (Table 1). The pronounced biological activity of these substances at a low concentration (10^−7^ M) in the medium was presumably a result of their regulatory signaling role in the cell.

Gly had a less pronounced growth stimulating effect the dipeptides (Table 1).

Compared to the control, Gly increased the total length of seedlings and the length of the main root increased by about 10–20% (Table 1). However, the appearance of lateral roots indicates that Gly, like dipeptides, stimulated the development of the tobacco root system.

Thus, the stimulating effect of Gly and dipeptides was accompanied by the appearance of additional roots and greater seedling biomass compared with the control.

### 2.2. Fluorescence and Light Microscopy

The results revealed zoning of root staining (Figure 2). In the absence of FITC or FITC-labeled peptides, the roots showed no autofluorescence (Figure 2a–c). When incubated in the presence of FITC-labeled Gly, GlyGly, and GlyAsp, the root cells showed fluorescence in different root zones and cell compartments: in the cell wall, cytoplasm, and nucleus (Figure 2).

When treated with FITC-Gly, diffused staining was observed in the root cap and meristem and intense staining was observed in the elongation zone (Figure 2h), mainly in the cellular cytoplasm (Figure 2i). FITC-GlyGly, when penetrating in the cap zone, forms diffuse staining and is more intense in the meristem zone (Figure 2k). In this case, intense staining of the cell wall, cytoplasm, and nuclei is observed (Figure 2l). A particularly intense fluorescence was observed in the meristem when treated with FITC-GlyAsp (Figure 2n), with the most intense staining in the nucleus (Figure 2o). This indicates that FITC-GlyAsp localize to a greater extent in the nucleus than in the cytoplasm.

Thus, FITC-labeled Gly and dipeptides stained the cell walls, cytoplasm, and nuclei in root cells, although the intensity of staining differed among cells. However, certain areas of the root showed no stained cells.

The localization pattern of the peptides indicates that the FITC-conjugated peptides penetrate and accumulate in tobacco roots. However, the nature of the accumulation of these substances varies among the root tissues and cells. Thus, the appearance and accumulation of the studied peptides in the root is tissue-specific. This may be because cells in different root zones exhibit different competence levels for interaction, penetration and accumulation of peptides. Thus, the determination of the localization of peptides can be effectively used for the rapid assessment of the status of root cells under different growth conditions and under the influence of various factors.

### 2.3. Root Hairs

Since the elongation and root hair zones are mainly responsible for the absorption of nutrients, we investigated the intact parts of the tobacco root (Figure 3). The root hairs, located in the root absorption zone, create a large surface for the absorption of water and nutrients. In the presence of peptides, root hairs varied in length, density, and pubescence pattern (Figure 3). Compared with the control, root hairs were 2-, 4-, and 6-fold longer in the presence of Gly, GlyGly, and GlyAsp, respectively (Figure 3). Additionally, root hairs treated with GlyAsp were more branched than those treated with other peptides (Figure 3).

### 2.4. Expression of Genes Controlling Root Hair Growth

In plants, several systems of regulation levels (active complexes) presumably control root hair growth. In this study, we considered two active complexes. One of these complexes includes six genes (*CPC*, *ETC*, *GL2*, *GL3/EGL3*, *TTG*, and *WER*), three of which (*CPC*, *WER*, and *GL2*) are expressed in H cells, along with *bHLH*, which acts as a linker. The *WER* gene suppresses the expression of *CPC* and *GL2* genes in H cells, thereby initiating the formation and growth of root hairs. The other complex includes *WER*, *CPC*, and *GTL1* genes, where the latter inhibits the growth of root hairs.

The bHLH54 and bHLH66 transcription factors activate the expression of genes involved in the formation of root hairs, such as *CPC* and *WER*. Overexpression of the *bHLH54* gene leads to the formation of longer root hairs, while the *bHLH66* gene controls the differentiation of root hairs. Moreover, the *bHLH66*-dependent pathway shows a strong association with root hair branching.

The expression of the *bHLH66* gene in hairless cells (N cells) of tobacco roots was unaffected by Gly and GlyGly compared with the control. By contrast, treatment with GlyAsp increased the expression of *bHLH66* by more than 2-fold in comparison with the control (Figure 4). Additionally, root hairs treated with GlyAsp showed a more branched structure than those treated with Gly and GlyGly (Figure 3).

Gly and dipeptides increased the expression of *bHLH54* by 1.5–1.7-fold compared with the control (Figure 4), consistent with their effect on root hair length (Figure 3). However, between the data shown in Figure 3 and Figure 4, there is no clear dependence, i.e., the levels of *bHLH54* expression in tobacco grown in the presence of both Gly and GlyGly, and GlyAsp are close to each other, and the length of the hairs differs significantly between them.

Compared with the control, Gly and GlyGly, increased the expression of the *WER* gene, which initiates the formation of root hairs, by 2-fold, whereas GlyAsp increased *WER* expression by ~3-fold. However, opposite results were obtained for the CPC gene, which is also involved in root hair initiation. The relative expression level of the *CPC* gene was increased by more than 3-fold in the presence of Gly and GlyGly, and by only 2-fold in the presence of GlyAsp.

However, these data do not explain the difference in the length of root hairs among tobacco grown in the presence of Gly, GlyGly, and GlyAsp, since the elongation of root hairs was observed in all treatments compared with the control (Figure 3).

In plants, there is a system for regulating the length of root hairs. The same cells contain genes that activate hair growth and those that inhibit hair growth. These genes are expressed in the cell simultaneously; however, activation of some genes and inhibition other genes are observed at different root hair developmental stages and under the influence of various factors.

Gly and both dipeptides inhibited the expression of *GTL1* and *GL2*, which suppress the growth of root hairs (Figure 4). However, if the suppression of *GL2* was insignificant compared with the control (and in the case of GlyAsp there was a slight increase), then the expression level of *GTL1* was 1.5–2.5-fold lower than the control. This was especially true for tobacco seedlings grown in the presence of Gly and GlyGly.

The activation of *WER*, *CPC*, *bHLH54*, and *bHLH66* genes and the inhibition of *GTL1* and *GL2* genes could probably explain the lengthening of the root hairs in tobacco seedlings grown in the presence of Gly, GlyGly, and GlyAsp compared with the control.

Thus, despite the similar structure of peptides, their impact on root hair growth differed, most likely because the neutral Gly amino acid and GlyGly dipeptide and the negatively charged GlyAsp dipeptide bind to different motifs in functionally important proteins. Genes expression levels differed slightly between tobacco plants grown in the presence of GlyGly and Gly, most likely because of differences in the binding constants between them and protein motifs.

## 3. Discussion

Short peptides, such as GlyGly and GlyAsp, and the amino acid Gly, present in the soil as waste products of soil microorganisms, penetrate the plant roots. Experiments with FITC-labeled additives showed fluorescence in the root cap apex, root meristem and extension and absorption zones. FITC-labeled peptides and Gly penetrated into the plant cell and localized to different subcellular compartments, including the nucleus, cytoplasm and cell wall. Penetration of these compounds into the nucleus can be accompanied by a change in chromatin structure. 

Nutrients and water are transported through the root hairs and differentiating vessels, and are distributed throughout the plant. Plants absorb water from the soil by osmosis. Root hair cells are adapted to osmosis, as they have a large surface area and a large permanent vacuole.

The pronounced biological activity of these GlyGly and GlyAsp, and the amino acid Gly at such a low concentration in the medium (10^−7^ M) is presumably due to the fact that they perform a regulatory signaling function in the cell like peptide hormones.

For many years, our knowledge of intercellular signaling in plants was limited to only five hormones—auxin, cytokinin, ethylene, gibberellin, and abscisic acid. However, in recent years it has been proven that secreted peptides form a new class of signaling molecules. Peptide hormones play a key role in many physiological processes, coordinating developmental and environmental signals between different cells. Small peptides are important signaling molecules that coordinate comprehensive cell-to-cell communication in many aspects of plant development [38,39,40,41,42]. Hormonal peptides play a critical role in plant growth and development, including defense mechanisms in response to pest damage, control of cell division and expansion, and pollen incompatibility [44].

The first plant peptide that was shown to have signaling activity was systemin, consisting of 18 amino acid residues, was isolated from tomato (*Lycopersicon esculentum*) [45]. It has been identified as a component of the systemic response to wounds following mechanical injury by insect pests. To date, several classes of functionally active peptides have been identified. However, due to the small size of peptide molecules and their cDNAs, identification of the functions of peptides in plants presents a certain difficulty. The family of peptides (CLAVATA/CLE) in plants has been identified which play a universal role in the development of such processes associated with the restoration of stem cells [46]. The CLE peptides negatively regulate the development of the meristem. Overexpression of CLE40 leads to inhibition of the growth of the root system [47].

Another family of hormonal peptides that affect the development of the root system are peptides containing sulfated tyrosine [48]. It is a family of peptides called root growth factor, RGF, which are required to maintain the root stem cell niche and cell proliferation, and which positively regulate the development of the meristem [49]. The branched root systems, which are formed as a result of the formation of lateral roots, are necessary to absorb water and nutrients from the soil and to firmly anchor the plant in the soil. It was shown that the development of lateral root roots depends on auxin. However, it has been found that the peptides CLEL6/RGF6 and CLEL7/RGF5 can regulate cell division of pericycle cells for lateral root initiation independently of auxin [50].

The secreted peptide RALF (fast alkalinization factor) with 5 kDa, has the ability to increase the pH of the medium in cultured cells [51]. The RALF peptide is most strongly expressed in roots and its participation in root development has been suggested. When it interacts with a receptor that includes the interaction of the peptide with the FERONIA receptor (FER), a complex is formed that helps to suppress the elongation of root cells. Recent studies have shown that the RALF1-FER complex is able, through activation, to regulate the synthesis of root hair proteins, including RSL4. RALF1-FER-RSL4 regulates root hair cell size by interacting with auxin and activating transcription [52].

Peptide hormones are the main regulators in plants, they participate in many aspects of the plant life cycle, including development and environmental responses, similar to the functions of canonical phytohormones. While the effect of exogenous peptides on animals has been studied for several years [53,54], the effect of exogenous peptides on plants is still poorly understood. 

Although the molecular mechanisms underlying the effect of exogenous peptides on cellular processes remain poorly understood, one of the possible mechanisms is the regulation of gene transcription. GlyGly, GlyAsp, and Gly, for example, can bind to regulatory proteins to influence gene expression. These compounds act at low concentrations (10^−7^ M) in this study, suggesting that the regulation of gene expression by short peptides is most likely of an epigenetic nature. A previous study showed that short exogenous peptides (GlyGly and GlyAsp) and Gly selectively alter the expression of genes encoding DNA methyltransferases and SNF2 proteins [55]. All dipeptides and free Gly reduced the expression of the *DRM2* methyltransferase gene, which carries out de novo DNA methylation, by 2–4-fold.

In most cases, DNA methylation is accompanied by gene silencing and, therefore, a decrease in DNA methylation level is accompanied by an increase in gene expression. A change in the level of DNA methylation in the presence of dipeptides and Gly can occur either because of the suppression of the expression of genes encoding DNA methyltransferases or because these compounds bind directly to DNA, thereby blocking its methylation. However, a change in the methylation level can also occur as a result of a change in the accessibility of specific DNA sequences to enzymes. Access to the genome can be altered either by covalent modifications of histones or by chromatin remodeling, altering the non-covalent interactions between DNA and histones [56,57], thus providing important epigenetic mechanisms for regulating gene expression [58]. ATP-related changes in the organization of nucleosomes by ATPases, belonging to the SNF2 family of proteins, represent a significant part of the chromatin remodeling activity [59]. It has been shown that Gly, GlyGly, and GlyAsp modulate the expression of genes encoding SNF2 family proteins, decreasing the availability of DNA for methylation, especially for *de novo* methylation [55]. Peptides can interact with the N- and C-termini of histones in the chromatin, blocking their enzymatic modification [60].

## 4. Materials and Methods

### 4.1. Plant Material

Seeds of tobacco (*Nicotiana tabacum* L.) cultivar Samsun were placed in flasks containing hormone-free Murashige–Skoog (MS) agar medium supplemented with or without 10^−7^ M Gly, GlyGly, and GlyAsp. Experiments were carried out in four replicates. After 28 days, the seedling fresh weight, seedling height, and root and seedling length were evaluated. The last parameter was measured under an Olympus BX51 microscope (Tokyo, Japan) furnished with the Cell program. The calculation of the main statistical parameters was carried out according to standard methods, and Statistica 6.0 and STATAN programs for statistical data processing were used.

### 4.2. Preparation of FITC-Labeled Peptides

A solution of FITC (1 μg in 10 μL of 0.5 M sodium bicarbonate) was added to a solution of Gly, GlyGly, and GlyAsp in 0.01 M Tris-HCl buffer (pH 7.0) at a ratio of 1.2:1 [61]. The reaction mixture was incubated at room temperature for 30 min with constant shaking. The obtained fluorescently labeled peptides were analyzed by chromatography on a BioLogic DuoFlow chromatograph on a C-18 column in a concentration gradient of acetonitrile (0–100%) containing 1% trifluoroacetic acid.

### 4.3. Fluorescence Microscopy

To determine the localization of peptides in root cells, the FITC-labeled dipeptides and Gly were used at a concentration of 10^−5^ M; control FITC and FITC without peptides were used as controls [62]. The incubation time was 20 h. Root tips (4–5 mm) were excised from tobacco seedling roots, and fixed in 4% paraformaldehyde (Sigma-Aldrich, St. Louis, USA) for 1.5 h at room temperature. After three washes with phosphate-buffered saline (PBS), the root tips were mounted on glass slides and embedded in Moviol. Fluorescence was analyzed at a wavelength of 490 nm using Olympus BX51 microscope (Tokyo, Japan) with 10× and 20× objective lens. Photographs were obtained using Color View digital camera (Munster, Germany) [63].

### 4.4. Light Microscopy

Roots stained with FITC-labeled peptides were fixed in 4% paraformaldehyde and embedded in LRW, as described previously [53]. Then, semi-thin root sections (1000 μm) were obtained using the LKB-III microtome (LKB, d procedure. For TEM, the embedded samples were sectioned using an ultramicrotome LKB-III (LKB, Bromm Sweden). Fluorescence was analyzed at a wavelength of 490 nm using Olympus BX51 microscope (Tokyo, Japan) with 10× and 20× objective lens. Photographs were captured using Color View digital camera (Munster, Germany). Photographs of root hairs and determination of their length along the main root were taken with a digital camera Rising View (Beijing, China). 

### 4.5. RNA Isolation

Total RNA was isolated from tobacco roots, from which root hairs were carefully removed, using RNA-Extran RNA isolation reagent kit (Syntol, Moscow, Russia), according to the manufacturer’s instructions. The concentration of isolated RNA preparations was determined on NanoPhotometer IMPLEN.

Then, cDNA was obtained using a set of reverse transcription reagents (Syntol, Moscow, Russia), according to the manufacturer’s instructions.

### 4.6. Real-Time PCR

Information on the primary structure of *Nicotiana tabacum* genes was obtained from the National Center for Biotechnology Information (NCBI). Primers for these genes were designed using the NCBI Primer-BLAST web tool, and synthesized at Syntol. Real-time PCR was performed on CFX96 Touch Real-Time PCR Detection System (Bio-Rad) using a set of RT-PCR reagents and SYBR Green (Syntol, Moscow, Russia). The relative expression level of genes was calculated using a calibration curve constructed with PCR products derived from the *GAPDH* gene. The efficiency of real-time PCR was calculated as E% = (10^−1^/s^−1^) − 100, where s is the slope of the dependence of the decimal logarithm of C_t_ values on the cDNA concentration. The efficiency of RT-PCR performed using primers corresponding to the genes of interest was 95–96%. The calculation of the main statistical parameters was carried out according to STATAN program on a significance level *p* < 0.05.

The equipment of the Center for Collective Use of the Federal State Budgetary Scientific Institution VNIISB RAS was used in this study.

## 5. Conclusions

Nutrients and water are carried through the root hairs and root differentiating vessels and are distributed throughout the plant. The ability to regulate root hair growth can have a direct impact on sustainable organic farming through improved water and nutrient intake.

Short peptides GlyGly and GlyAsp and Gly penetrate into the roots and their accumulation is tissue-specific, with predominant localization in the root cap, meristem, zones of extension and absorption. Peptides penetrate into the epidermal and cortical cells and show preferential localization in the nucleus than in the cytoplasm.

Dipeptides and Gly have a marked effect on the development and elongation of tobacco root hairs. In the presence of peptides, the root hairs were longer, denser, and differed in the character of pubescence in comparison with the control. 

The mechanism of the action of dipeptides and Gly differed, depending on their structure and charge. Overall, the dipeptides GlyGly and GlyAsp and the amino acid Gly showed pronounced physiological activity, and can be considered as positive regulators of the formation of root hairs.

It is suggested that GlyGly, GlyAsp, and Gly are a new class of plant growth and development regulators. The most studied mechanism of regulation of genome transcription is epigenetic regulation, which includes DNA methylation and modification of histone proteins.

## Figures and Tables

**Figure 1 plants-11-00852-f001:**
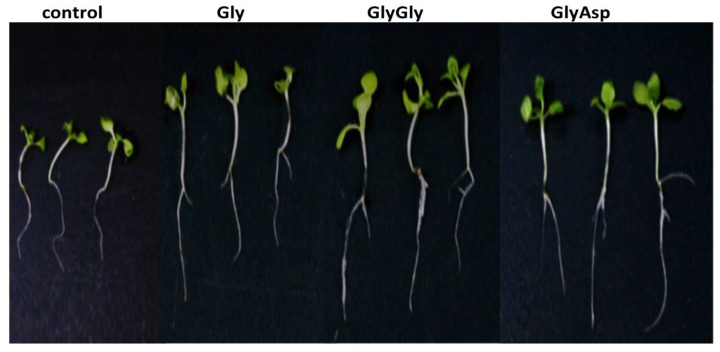
Tobacco seedlings grown on Murashige and Skoog (MS) medium supplemented with or without Gly amino acid and GlyGly and GlyAsp dipeptides.

**Figure 2 plants-11-00852-f002:**
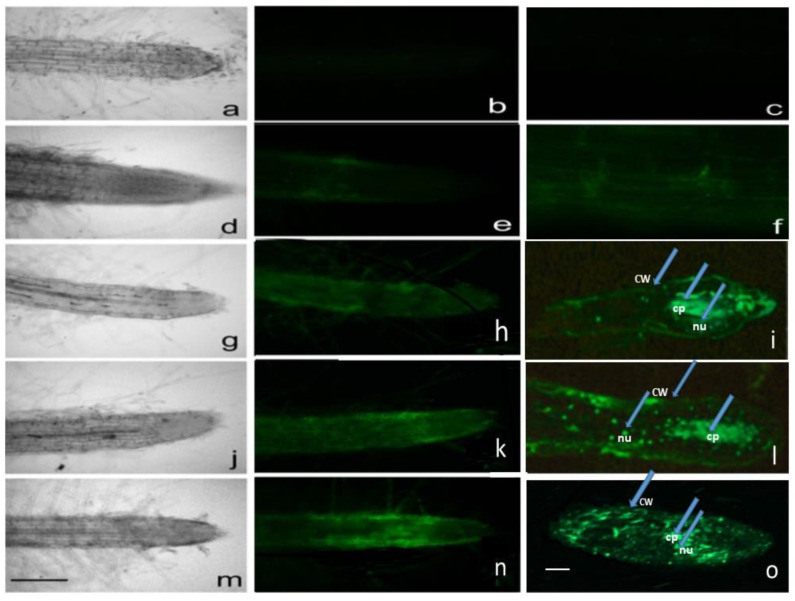
Visualization of FITC-labeled Gly and dipeptides in tobacco seedling roots. (**a**–**c**) Control roots. (**d**–**f**) Roots incubated with FITC. (**g**–**i**) Roots incubated with FITC-Gly. (**j**–**l**) Roots incubated with FITC-GlyGly. (**m**–**o**) Roots incubated with FITC-GlyAsp. Scale = 200 µm. (**c**,**f**,**i**,**l**,**o**) Semi-thin root sections. Arrows point to cw—cell wall; cp—cytoplasm, nu—nuclei.

**Figure 3 plants-11-00852-f003:**
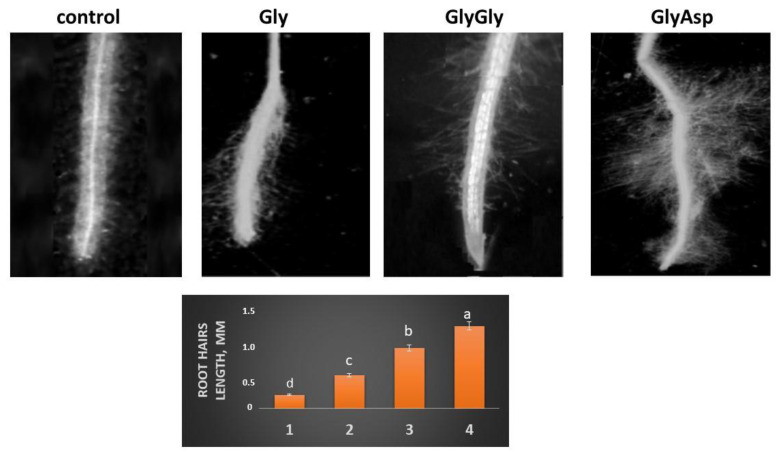
Effect of peptides on the length, density, and pubescence of root hairs of tobacco seedlings. 1—control; 2—Gly; 3—GlyGly; 4—GlyAsp. Scale = 1000 µm. The mean values (*n* = 20) and their standard deviations are shown according to significant value, *p* < 0.05. Different letters above the bars indicate significantly different values.

**Figure 4 plants-11-00852-f004:**
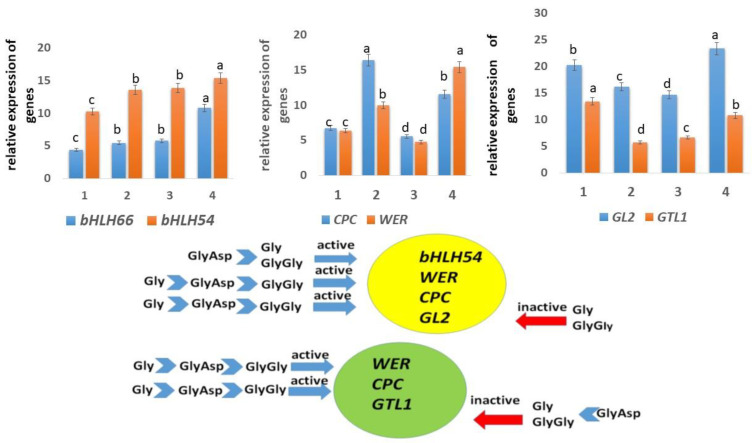
Relative expression levels of genes in tobacco seedling roots. 1—control; 2—Gly; 3—GlyGly; and 4—GlyAsp. Data represent mean ± standard deviation. The diagram below indicates the activation or repression of genes in the presence of Gly and dipeptides. Standard deviations are shown according to significant value, *p* < 0.05. Different letters above the bars indicate significantly different values.

**Table 1 plants-11-00852-t001:** Influence of Gly, GlyGly, and GlyAsp on the morphometric parameters of tobacco seedlings.

Additives	Mean ± Standard Error (*n* = 20)		
	Seedling Fresh Weight (mg)	Total Seedling Length (mm)	Main Root Length (mm)
Control	0.412 ± 0.03 d	21.6 ± 1.2 b	12.4 ± 0.6 c
Gly	0.440 ± 0.03 c	23.7 ± 1.6 a	14.8 ± 1.1 b
GlyGly	0.587 ± 0.04 a	24.5 ± 1.6 a	17.9 ± 0.8 a
GlyAsp	0.561 ± 0.04 b	25.0 ± 1.5 a	17.8 ± 1.0 a

Standard deviations are shown according to significant value, *p* < 0.05. Different letters indicate significantly different values.

## Data Availability

Not applicable.

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
