# Peer review of "Short Peptides Induce Development of Root Hair Nicotiana tabacum"

_plants, 2022, doi:10.3390/plants11070852_

Round 1

Reviewer 1 Report

Comments

This article reports the enhancing effect of Gly and two dipeptides, GlyGlu and GlyAsp on the growth of root hairs of tobacco roots.

My main critique is that in order to prove the specificity of the effects by the studied peptides controls of another amino acid and di-peptide should have been included in the experiments.

The remarks under Table 1 and Fig. 4 mention “Student criterion, p<0.05.” I guess they mean that a Student’s t-test was used to determine the significance of the differences among the data in each group. In cases like these, a multiple-range test such as Tukey's-range-test should have been used.

Looking at Fig. 2 I am not convinced that the interpretation in the two paragraphs above it is accurate. They should have at least provided light-microscope images at the magnification of the images in the right column.

The discussion contains several topics that are remotely associated with the results of the reported experiment, for example, see lines 295-306 that discuss the effects of peptides that do not penetrate the cells.

The reference to “sustainable organic farming” (Line 395) is farfetched since the fate of such peptides at such a low concentration in the soil is dubious.

Linguistics and technical

Line 7 – What do they mean by “Root hairs …reduce environmental effects …”?

Line 12 – By “stretch zone” do they mean “elongation zone”?

Line 17 – Replace ”differ” with “different”

Line 30 – Replace “cylinders” with “root sheaths”

Line 32 – Replace “resistance” with “adaptation”

Line 55 – Replace “linked” with“link”

Lines 63-64 – Replace “and directly involved” with“and is directly involved”

Line 72 – Replace “issue” with “question”

Line 97 – Replace “are” with “were”

Line 101 – Relace “from” with“of”

Line 127 – Move the table caption and header to the next page

Line 228 – Replace “development” with “developmental”

Line 236 – Replace “… be explained by …” with “explain”

Line 413 – replace “light” with “light microscopy”

I did not check all the references but noticed that:

Not all article titles are in “Sentence case” but in some, all words are capitalized.

 Line 428 – They should add “Brown, L.K.;” after “Haling, R.E.;”

Line 452 – Delete an extra semicolon

Line 458 – Change “/” to ”.”

Line 470 – Arabidopsis thaliana should be in Italics

Lines 520, 532 – Remove underlines

Make sure all semicolons are followed by a space

Author Response

Thank you for your valuable comments. We tried to fix everything.

Reviewer 2 Report

Dear Authors

Please: 1. find the attached revised manuscript and give responses to the comments.

2. provide another photo in Figure 2 to explain the results well

3. please follow the Journal rules in the References

Author Response

We are very grateful for your valuable comments.

Round 2

Reviewer 2 Report

The manuscript was improved

This manuscript is a resubmission of an earlier submission. The following is a list of the peer review reports and author responses from that submission.

Round 1

Reviewer 1 Report

The subject of the manuscript is consistent with the scope of the Journal. The keywords correspond well to the scope of the research.

The aim of this study was to investigate the effect of glycine (Gly) amino acid and 94 GlyGly and Gly-aspartic acid (GlyAsp) dipeptides on the regulation of root hair growth 95 in tobacco (Nicotiana tabacum L.).

I think the paper needs some corrections:

1) add chemical composition of agar medium (if you have),

2) add more detailed information about experiment (it is only some basic information now),

3) add detailed information about used statistical methods,

4) add table numbers (lines: 108, 116, 117 and 119),

5) the discussion is short (only 6 references were cited), add some new references to Discussion section,

6) standardize References section.

Paper needs some editorial corrections (see: Instructions for Authors).  

You must check your paper very exactly and correct all mistakes in the paper.

The manuscript entitled “Short peptides induce development of Root Hair Nicotiana tabacum”, amended according to the revision, can be accepted for publishing in Plants.

Author Response

Thank you very much for your review and valuable comments.
1. We used a standard hormone-free set of Murashige - Skoog.
2. In the articles, we refer to the conditions of the experiment, which are described in detail in the links provided.
3. We added the programs that we used for statistical processing to the experimental part.
4. There is only one table in our manuscript. However, we have numbered it.
5. We have added links to the discussion. There are no data in the literature on the effect of exogenous short peptides on plants. The effect of exogenous peptides on animals is now being intensively studied. But our short peptides act on root hairs, so we did not compare them with animal cells.
6. Sorry, link misalignment occurred during the submission process.

Reviewer 2 Report

The study of Fedoreyeva et al. focused on effects of different types of short peptides on inducing root hair development.

1) Full names of all short names should be explained in the Abstract.

2) In the introduction section, author used too many paragraphs to explain the development of roots and root hairs, but only used one short paragraph to demonstrate the physiological function of peptides in plants. Please add more contents about the physiological function of peptodes.

3) No significant analysis was observed in the Table 1. Please add significant analysis as marked by using different lower-case letters. In the Fig. 4 and 5, significant identification as reflected by different lower-case letters should be also add  to make these figures look clearer.

4) References and citations should not be put in the results section.

5) The quality of Fig. 4 is really bad and not good for publication. Please revise it carefully! What is the meaning of number 1, 2,3 and 4 below the bottom of blue and orange columns? Fig. 4 is really hard to understand!

6) The discussion section was too simple too elucidate the effects of different types of peptides on improving root hairs, which mean the discussion section is weak and a lack of readability and scientificalness. 

Author Response

Thank you very much for your review and valuable comments.
1. We added the full name of the peptides to Abstract
2 We have added to the introduction section a description of the physiological function of endogenous peptides in plants. Data on the physiological function of exogenous peptides in plants are under study.
3. Thanks for your comment. We have entered significant criteria in the table and in the figures. 
4. Sorry, we decided to leave links in the Results section.
5. We tried to improve the quality of figure 4.
6. In Discussion section we added some additions. The results on the effect of exogenous peptides on plants are limited (only our studies). The work presents data only on the effect of exogenous peptides on the development of root hairs. We did not dare to discuss in detail the still poorly understood mechanism of their action.

Reviewer 3 Report

The manuscript by Fedoreyeva et al. ("Short Peptides Induce Development of Root Hair Nicotiana tabacum") reports on the growth promoting effects of an amino acid (Gly) and two dipeptides (Gly-Gly and Gly-Asp) when added to seedlings' growth medium at very low concentrations. These molecules appear to promote seedling fresh weight, total seedling length and main root elongation relative to controls. Shoot height is slightly decreased in the presence of the dipeptides relative to controls. Furthermore, these compounds also promote root branching and root hair length as well as hair branching (for GlyAsp). Treatments with FITC-labeled compounds indicate differential uptake by different tissues of the root tip, and also signal accumulation in the nucleus and cytoplasm of these cells. Finally, expression studies indicate treatments with these signaling molecules result in differential activation of expression of target genes known to encode transcription factors that contribute as activators and/or inhibitors of root hair initiation and/or development. Overall, the data are used to suggest a role for these molecules in modulating "epidermal cell differentiation and N-to-H cell conversion".

While the data are very interesting and suggestive of an important role for these molecules in modulating plant morphology, growth and cellular differentiation, I strongly believe the data are too preliminary to warrant publication at this time, for the following reasons:

  1. It is totally unclear why the authors initially chose to characterize the effect of these three molecules (Gly, Gly-Gly and Gly-Asp) on plant growth and development as opposed to any other amino acid or dipeptide molecules. The rationale behind these experiments is simply not documented. Would any other amino acid or dipeptide have similar roles in plant development? Are Gly, Gly-Gly and Gly-Asp commonly found in root communities/soils? At what concentrations?
  2. Many of the quantified parameters are poorly defined, and comparative analyses are completely devoid of statistical analyses of significance. Good examples of the lack of statistical analyses of the data are shown in Table 1 and Figure 3.
  3. The authors mention on page 3 that the area of the leaf blade of dipeptide-treated seedlings is 1.5-fold higher compared to control seedlings. Those data are not shown in the table or anywhere else in the manuscript, though!
  4. The localization studies rely on the use of FITC-labeled amino acid or dipeptide molecules. However, the authors never discuss the possibility that the signals observed in these experiments might not be associated with the FITC-labeled Gly or dipeptide. In fact, this signal could very well be derived from a metabolic product of these molecules.
  5. Similarly, to be more meaningful, these localization studies should be accompanied by control experiments investigating the effect of these FITC-labeled molecules on seedling growth, root branching and root hairs, to verify the labeling did not alter the function of these molecules.
  6. The impact of Gly, Gly-Gly and Gly-Asp on root hairs seems quite striking. However, the quantification of these effects is rather limited. A quick look at figure 3 indicates a rather heterogeneous distribution of root hair lengths in dipeptide-treated roots relative to control. Where were root hair lengths measured along the roots / lateral roots in dipeptide-treated samples relative to controls to generate the data illustrated in the lower panel of Figure 3? More importantly, the authors also mention that more root hairs are branched in Gly-Asp-treated roots relative to control. They refer to figure 3 to illustrate this point. Unfortunately, the pictures shown in figure 3 are of very poor resolution, preventing us from evaluating this phenotype.
  7. The authors mention in their conclusion (page 10) that the dipeptides can be considered as "regulators of epidermal cell differentiation and N-to-H cell conversion". None of the data shown in this manuscript allow us to evaluate the existence of N-to-H conversion. Here again, better micrographs should be provided showing close-up examples of root hairs developing ectopically in cells that should normally develop as atrichoblasts.
  8. The effect of Gly, Gly-Gly and Gly-Asp on the expression of transcription factor genes previously implicated in root hair development regulation is documented in Figure 4. While very interesting, these data are not accompanied by statistical analyses of significance. Therefore, it is impossible to evaluate the significance of observed differences. Furthermore, no information is provided defining the numbers of biological and technical replicates included in this analysis. Finally, for many of these transcription factor genes, the cellular distribution of their expression matters a lot in defining their contribution to root-hair development. The compounds could alter their tissue specificity of expression, thereby affecting root hair development in a different manner than discussed in this manuscript.
  9. Additional minor suggestions:
  10. The lower panel of Figure 1 does not add much to the figure. This figure could and should be improved by increasing the contrast to better illustrate root morphology;
  11. In Table 1, "Seedling Height (mm)" should probably be replaced by "Shoot height (mm)" (because a seedling contains both cotyledons, hypocotyls (shoots) and roots).

Overall, this manuscript describes very exciting preliminary data supporting differential roles for Gly, Gly-Gly and Gly-Asp in the regulation of seedling growth, morphology, root branching and root hair formation. However, the lack of statistical analysis of the data, the suboptimal quality of photographs not allowing interpretation of the reported data, and a lack of appropriate controls in the analysis of compound localization, all dampen my enthusiasm for its publication under this form.

Author Response

We thank you for carefully reading our manuscript, valuable comments and suggestions. We have tried to respond to your comments. And in future research, we will certainly take into account your suggestions.

Round 2

Reviewer 1 Report

Dear authors,

This is very bad that you can not find more references related to this topic to discussion section.
If the editor will decide so, this manuscript can be published in Plants.

Author Response

We thank you for your recommendation to publish our study

Reviewer 2 Report

All my suggestions were not solved and revised by authors. The article has serious flaws and additional experiments and analysis needed.

Author Response

We thank you for your valuable comments. We've significantly revamped the discussion and added links. We have increased the section on materials and methods, especially statistical processing

Reviewer 3 Report

This revised version of the manuscript by Larisa I. Fedoreyeva et al. (Short Peptides Induce Development of Root Hair in Nicotiana tabacum") addresses some of the points I raised in my review of the first draft, but several important suggestions were not followed.

In answer to the request of additional information justifying the study of these compounds in the regulation of seedling growth and root hair development, the authors added a satisfactory discussion of previously reported data documenting such effects. They also expanded the discussion to better evaluate the biological significance of their results. They removed an unnecessary panel in figure 1, and corrected their nomenclature associated with distinct seedling parts in Table 1.

On the other hand, several important suggestions/criticisms were not properly addressed in this revised draft:

1) The statistical analysis of the data is not properly addressed. They mention that "the calculation was carried out according to Student's criterion". However, they never show which samples are statistically significantly different from the controls in their experiments (being Table 1, Figure 3 or Figure 4). Each one of those datasets should include a symbol that identifies those that are considered significantly different from controls using the t-test. For instance, is seedling fresh weight in Gly-treated samples significantly different from control? Is shoot height of Gly-treated seedlings significantly different from control? All treatments and quantitative parameters should be treated this way. Similarly, in Figure 4, does bHLH66 expression in conditions "2" and "3" significantly different from the control (which I assume to be "1", but do not know for sure because this has not been defined in the legend). Does GTL1 expression in condition 4 differ significantly between condition "4" and the control? How about expression of CPC and WER under condition 3 relative to control? Here again, all comparisons should be evaluated similarly.

2) The authors mention having removed sentences relating to leaf-blade area in their revisions. In fact, one sentence describing this precise parameter can be found in Page 4, line 131.

3) The potential artifacts associated with relying on FITC-labeled peptides to identify root-tip cells targeted by these molecules have not been addressed at all in the manuscript. In their response to reviewers, the authors claim having verified penetration of the labeled compound into human HELA cells. They claim that they "did not consider the issue of metabolism since FITC is not a natural component and it is not metabolized in the body". My answer to this comment is that many unnatural compounds are metabolized by plants. They also claim "It can only be assumed that the bond between FITC and the peptide can be hydrolyzed in the body. But it doesn't matter, since the peptide has already entered the plants". My answer to this comment is that it matters a lot if they want to claim that the peptides are targeted in different cellular compartments within the plant such as nucleus, cytoplasm and wall. To me, these data are not very meaningful in the absence of additional control experiments verifying that the labeled peptides remain intact in plant cells.

4) To the suggestion of testing the effect of these labeled peptides on the plant growth parameters investigated in this report, the authors argue they do not want to do this because "FITC would be toxic to plant development". This is a big red flag, and should be sufficient to discount the results of these labeling experiments altogether, if this is what they observe in reality.

5) To the suggestion of increasing resolution of the microphotographs shown in Figure 3 to illustrate root hair bifurcations, the authors claim replacing this panel by a new one with improved resolution. Unfortunately, this figure is still of insufficient resolution to illustrate this point. A close-up, unambiguous image should be added to this figure illustrating this point.

6) The authors do not address the basis for root hair density increase in this paper.

7) The discussion on the basis of altered expression of transcription factors in the root tip of peptide-treated vs control roots remains rather vague, not addressing possible changes in target gene expression patterns upon treatment.

Author Response

It was very kind of you to conduct such a detailed analysis of our manuscript. We are glad that you have appreciated the expansion of the Discussion section. We hope that all your criticisms were taken into account by us in the revised draft.

Round 3

Reviewer 2 Report

Current manuscript was not improved according to suggestions.

Author Response

Thank you again for reviewing the manuscript. 

We have reworked all the statistics and presented these results in a redesigned version.

Reviewer 3 Report

I do not think the suggestions made for improvement of the previous draft have been addressed in this revision. Below, I reply to the authors' replies to my previous comments:

  1. Reviewer Reply: The following sentence used several times in this document is worrisome when thinking about the statistical analysis of the data: "Standard deviations are shown according to Student's criterion, p<0.05. Different letters indicate 128 significantly different values." In reality, standard deviations and t tests are different things. Along those lines, the authors replied to the request for improved statistical analyses of significance by taking out of Table 1 the values of standard deviations for each measurement. It is important to both show the standard deviation values in this table (as shown in the previous draft), and show the results of t-analyses of significance. At the same time, t-tests involve pairwise comparisons between pairs of samples. For instance, in Table 1, first column, one would compare each treatment with the control, allowing identication of treatments leading to significantly different values (seedling fresh weight in this case) relative to control. A test that would allow identification of significantly different groups within multiple samples as indicated in revised Table 1 would be ANOVA (for instance). I would suggest including a statistician in the analysis of these data. The same comment holds for Figures 3 and 4.
  2. Reviewer Reply: Thank you.

  3. Reviewer Reply: Pretty much the same way as the amino acid or peptide attached to FITC modulates the cellular uptake of the molecule (none for FITC alone), FITC also has the potential of affecting uptake, localization and binding of the labeled molecule in plant cells (relative to Gly or the peptides). After all, FITC is actually larger than Gly or the peptides it is fused to in these molecules, calling for caution in the interpretation of localization studies with these molecules (and also receptor binding, as a matter of fact)! Assuming that peptide metabolism in plant cells is similar to that in animal cells is also a major assumption (after all, we know that key aspects of amino acid metabolism differ dramatically between animals and plants, such as biosynthesis, for instance)! In other words, without controls aimed at evaluating the validity of those assumptions, these localization experiments are prone to artifacts.

  4. Reviewer Reply: See reviewer reply to point 3.

  5. Reviewer Reply: This photo still fails to show root-hair bifurcation, I am afraid. It is also unfocused.

  6. Reviewer Reply: As discussed in my review of the first draft, I do not think this is adequately addressed.

  7. Reviewer Reply: Most of the transcription factors analyzed here control the specification of root hair cells vs atrichoblasts. Information on expression changes in response to gly and the peptides analyzed here would be very useful if careful analysis of root hair density were included in the study (see point 6), and the expression pattern of each differentially expressed gene were assessed. The data presented in this manuscript are preliminary and incomplete, at best.